# Structure of full-length ERGIC-53 in complex with MCFD2 for cargo transport

Satoshi Watanabe [1,2,3] ✉, Yoshiaki Kise [4], Kento Yonezawa [5,8], Mariko Inoue[1], Nobutaka Shimizu [5], Osamu Nureki [4] & Kenji Inaba [1,2,3,6,7] ✉

ERGIC-53 transports certain subsets of newly synthesized secretory proteins and membrane proteins from the endoplasmic reticulum to the Golgi apparatus. Despite numerous structural and functional studies since its identification, the overall architecture and mechanism of action of ERGIC-53 remain unclear. Here we present cryo-EM structures of full-length ERGIC-53 in complex with its functional partner MCFD2. These structures reveal that ERGIC-53 exists as a homotetramer, not a homohexamer as previously suggested, and comprises a four-leaf clover-like head and a long stalk composed of three sets of four-helix coiled-coil followed by a transmembrane domain. 3D variability analysis visualizes the flexible motion of the long stalk and local plasticity of the head region. Notably, MCFD2 is shown to possess a $Zn^{2+}$-binding site in its N-terminal lid, which appears to modulate cargo binding. Altogether, distinct mechanisms of cargo capture and release by ERGIC- 53 via the stalk bending and metal binding are proposed.

Secretory proteins and membrane proteins are newly synthesized in the endoplasmic reticulum (ER) and undergo oxidative folding and N-glycosylation with the help of luminal chaperones, glycosylation enzymes, and oxidoreductases[1,2]. A large portion of folded secretory proteins diffuse into the ER exit site, and are then transported into the Golgi apparatus (GA) by a bulk flow process[3]. However, subsets of secretory and membrane proteins are captured by cargo receptors in the ER, and subsequently transported to the GA in an accelerated manner[3].

ERGIC-53 (also named LMAN1, P58) is a cargo receptor that captures certain specific secretory proteins[4,5], including coagulation factor V (FV) and factor VIII (FVIII)[6,7], α1-antitrypsin[8–10], cathepsin C[11], cathepsin Z[12], Mac-2 binding protein[13], matrix metalloproteinase-9[14], and IgM[15]. Mutations of the *ergic-53* gene cause a genetic bleeding disorder called combined deficiency of FV and FVIII (F5F8D)[7,16]. Another causative gene of F5F8D is the multiple coagulation factor deficiency 2

gene (MCFD2)[16,17]. MCFD2 is a small protein composed of an EF-hand domain with two $Ca^{2+}$ binding sites, and functions as a binding partner of ERGIC-53 in a $Ca^{2+}$-dependent manner[17–20].

ERGIC-53 has also been reported to promote the ER-to-GA transport of the chaperone protein ERp44[21] and several membrane proteins such as γ-aminobutyric acid type A receptors (GABA$_A$Rs)[22] and NDST1[23]. Furthermore, ERGIC-53 interacts with surface glycoproteins of infected RNA viruses and enhances their viral propagation[24,25].

From the amino acid sequence, ERGIC-53 is predicted to be a single-pass transmembrane protein that consists of a luminal carbohydrate recognition domain (CRD), a long stalk domain, a transmembrane (TM) helix and a short cytoplasmic tail (CT)[4]. The CRD is structurally and functionally similar to a plant I-type lectin and binds to high-mannose type glycans of the target cargo proteins in a pH- and $Ca^{2+}$-dependent manner[11,18,19,26,27]. The CT contains a C-terminal KKFF motif required for the ERGIC-53 cycling between the ER and GA[28–30].

[1]Institute of Multidisciplinary Research for Advanced Materials, Tohoku University, Sendai, Miyagi 980-8577, Japan. [2]Department of Molecular and Chemical Life Sciences, Graduate School of Life Sciences, Tohoku University, Sendai, Miyagi 980-8577, Japan. [3]Department of Chemistry, Graduate School of Science, Tohoku University, Sendai, Miyagi 980-8578, Japan. [4]Department of Biological Sciences, Graduate School of Science, The University of Tokyo, Bunkyo-ku, Tokyo 113-0033, Japan. [5]Structural Biology Research Center, Institute of Materials Structure Science, High Energy Accelerator Research Organization (KEK), Tsukuba, Ibaraki 305-0801, Japan. [6]Medical Institute of Bioregulation, Kyushu University, Fukuoka 812-8582, Japan. [7]Core Research for Evolutional Science and Technology (CREST), Japan Agency for Medical Research and Development (AMED), Tokyo, Japan. [8]Present address: Center for Digital Green-innovation, Nara Institute of Science and Technology, Ikoma, Nara 630-0192, Japan. ✉e-mail: satoshi.watanabe.c1@tohoku.ac.jp; kenji.inaba.a1@tohoku.ac.jp

The stalk domain is predicted to be composed of four long α-helices followed by a long loop and the single-pass TM helix. The three stalk α-helices except the most N-terminal helix are predicted to form a coiled-coil[31]. The long loop between the fourth stalk helix and TM domain contains two conserved Cys residues (Cys466 and Cys475), which form intermolecular disulfide bonds[31].

After the discovery of ERGIC-53 in 1988, it was long believed that this cargo receptor existed as a covalent hexamer or covalent dimer via the interprotomer disulfide bonds between the stalk domain[32]. A subsequent study by Neve et al., however, proposed that ERGIC-53 existed in two forms, a covalent hexamer and a noncovalent hexamer, in which three of disulfide-bridged dimers of ERGIC-53 form the disulfide-linked hexamer (covalent hexamer) or are noncovalently assembled into the hexamer (noncovalent hexamer)[33]. This preceding study also demonstrated that the cysteine mutant that lacks the intermolecular disulfide bonds retained the hexameric state and was localized in the ERGIC as was wild-type ERGIC-53. It is thus suggested that the intermolecular disulfide bonds are not essential for the maintenance of its hexameric structure and cellular localization[33].

Although the oligomeric states of ERGIC-53 were analyzed by biochemical approaches, more detailed structural and functional studies have been focused on the CRD and its complex with MCFD2[19,34–38]. Therefore, it still remained elusive how full-length ERGIC-53 forms the putative hexamer and works as a cargo receptor in this oligomeric state. Here, we present cryo-electron microscopy (cryo-EM) structures of full-length ERGIC-53 in complex with MCFD2. Contrary to the previous reports, the present structures, in combination with size-exclusion chromatography combined with multiangle light scattering (SEC-MALS) and with small-angle X-ray scattering (SEC-SAXS) analysis, reveal a tetrameric rather than hexameric architecture, with a long flexible stalk domain. High-resolution structures of its headpiece elucidate the molecular basis of the tetramer formation of ERGIC-53 and the zinc ion (Zn²⁺)-binding to the N-terminal region of MCFD2 for likely regulation of cargo binding and release. Altogether, the present study provides insights into the mechanism of cargo transport by ERGIC-53 using the long flexible stalk domain and the cargo-binding head region regulated by $Zn^{2+}$.

## Results

### ERGIC-53 exists as a tetramer with a long extended structure

The previously proposed hexamer model of ERGIC-53 was based on PAGE analyses of crude purified samples[33]. To verify the oligomeric states of ERGIC-53 by more conclusive methods, we performed SEC-MALS analysis of purified full-length ERGIC-53. Recombinant full-length human ERGIC-53 was successfully purified to high purity from cumate-inducible stable HEK293T cell lines (Supplementary Fig. 1a). The monodisperse peak fractions from SEC elution contained two bands in nonreducing SDS-PAGE at ~300 kDa and ~110 kDa, which were presumed to result from the covalent and noncovalent hexamers of ERGIC-53, respectively. However, SEC-MALS conjugated analysis showed that the molecular masses of purified ERGIC-53 and its complex with MCFD2 were determined to be 230 and 280 kDa, respectively (Supplementary Fig. 1b–d). The determined molecular masses correspond to a homotetramer of ERGIC-53 alone (221.4 kDa) and a complex of four ERGIC-53 protomers with four MCFD2 molecules (277 kDa). The tetrameric state of ERGIC-53 was retained at weakly acidic Golgi pH (pH 6.5) (Supplementary Fig. 2). Thus, full-length ERGIC-53 likely exists as a mixture of covalent and noncovalent homotetramers, and each protomer forms a 1:1 complex with MCFD2. The hexamer model (332 kDa) suggested by preceding works did not receive further experimental support.

To gain further insight into the overall structure of ERGIC-53 in solution, we next performed SEC-SAXS analysis (Supplementary Fig. 1e and Supplementary Table 1). The pair distance distribution function P(r) of full-length ERGIC-53 showed two separate peaks and estimated its maximum molecular dimension ($D_{max}$) to be 347 Å, suggesting that this cargo receptor adopts an overall dumbbell-like shape with an unusually long length (Supplementary Fig. 1f, h). Compared with isolated ERGIC-53, the ERGIC-53-MCFD2 complex displayed almost the same $D_{max}$ and two slightly higher and sharper peaks in the P(r) function (Supplementary Fig. 1f, h), suggesting that in the complex, the corresponding two regions are more compact in conformation than those in isolated ERGIC-53. In the dimensionless Kratky plots of ERGIC-53 alone, the main peak was shifted to the larger $qR_g$ side (~3.5) and has a higher height (~2.5) than those of typical globular proteins, suggesting that the overall conformation of isolated ERGIC-53 is extended and in a highly asymmetric state[39]. On the other hand, the complex showed not only a slightly narrower width of the peak but also lower values at the high $qR_g$ region than ERGIC-53 alone, indicating that conformational flexibility or disorder was repressed upon complex formation with MCFD2 (Supplementary Fig. 1g). These results suggest that ERGIC-53 binds MCFD2 to adopt a more rigid or tightly folded structure than the isolated state.

### Cryo-EM structures of the head region of ERGIC-53 in complex with MCFD2

Based on the solution structure of the more rigid conformation of the ERGIC-53-MCFD2 complex, we performed cryo-EM analysis of full-length ERGIC-53 in complex with MCFD2. In the motion-corrected micrographs, long dumbbell-like particles of the complex were observed (Supplementary Fig. 3, red square), consistent with the P(r) function obtained in the SEC-SAXS analysis. However, conventional autoparticle picking programs failed to recognize the full-length ERGIC-53 particles due to their unusual shapes. Instead, the autopicking programs recognized two globular portions in the dumbbell as separate particles (green circles in Supplementary Fig. 3). 2D classification of these two globular portions of the full-length particles showed clear 2D average images of the N-terminal tetrameric head of the complex and the C-terminal portion composed of the TM domain (including detergent micelle) and a part of the stalk domain, respectively (Supplementary Fig. 4a). Consequently, we determined the cryo-EM structures of the head region of full-length ERGIC-53 complexed with MCFD2 in two forms (forms A and B) at 2.53 and 2.59 Å resolutions, respectively (Supplementary Fig. 4a–g and Supplementary Table 2). On the other hand, only low-resolution EM maps were obtained for the C-terminal portion, probably due to its high flexibility and heterogeneity in the shape of detergent micelles.

The present cryo-EM analysis reveals that the overall structure of the assembled head is composed of four ERGIC-53 protomers and four MCFD2 molecules, confirming that ERGIC-53 exists as the tetramer (Fig. 1a, b). Each head region consists of the CRD (residues 31–268), stalk helices 1 and 2 (S-H1 and S-H2), and loops connecting these domains (Fig. 1c, left). The tetrameric structure is stabilized by a central coiled-coil composed of the long S-H2 from each protomer (residues 326–367) (Fig. 1b and Supplementary Fig. 5a). The S-H2 forms a typical parallel four-helix coiled-coil. Hydrophobic residues are located at the heptad positions **a** and **d** in the coiled-coil, forming the hydrophobic core within it (Supplementary Fig. 5b–e). A polar residue Gln338 is also located in the **a2** layer, where water or some ion is bound to stabilize the polar interactions (Supplementary Fig. 5f). The local resolution of S-H1 and the loops between CRD and S-H1, and between S-H1 and S-H2 are relatively low, suggesting the high flexibility of these regions (Supplementary Fig. 4c). MCFD2 binds the CRD at a 1:1 molar ratio as observed in the crystal structures of the CRD-MCFD2 complex[34–37]. In addition, the S-H1 weakly associates with a part of MCFD2 (Fig. 1c and Supplementary Fig. 6a, inset 3), likely affecting the conformation of its N-terminal region (also see below).

### CRD head assembly

Although ERGIC-53 forms the homotetramer, the four CRDs are not related by a fourfold axis, but arranged in a C2 symmetry. Two CRDs

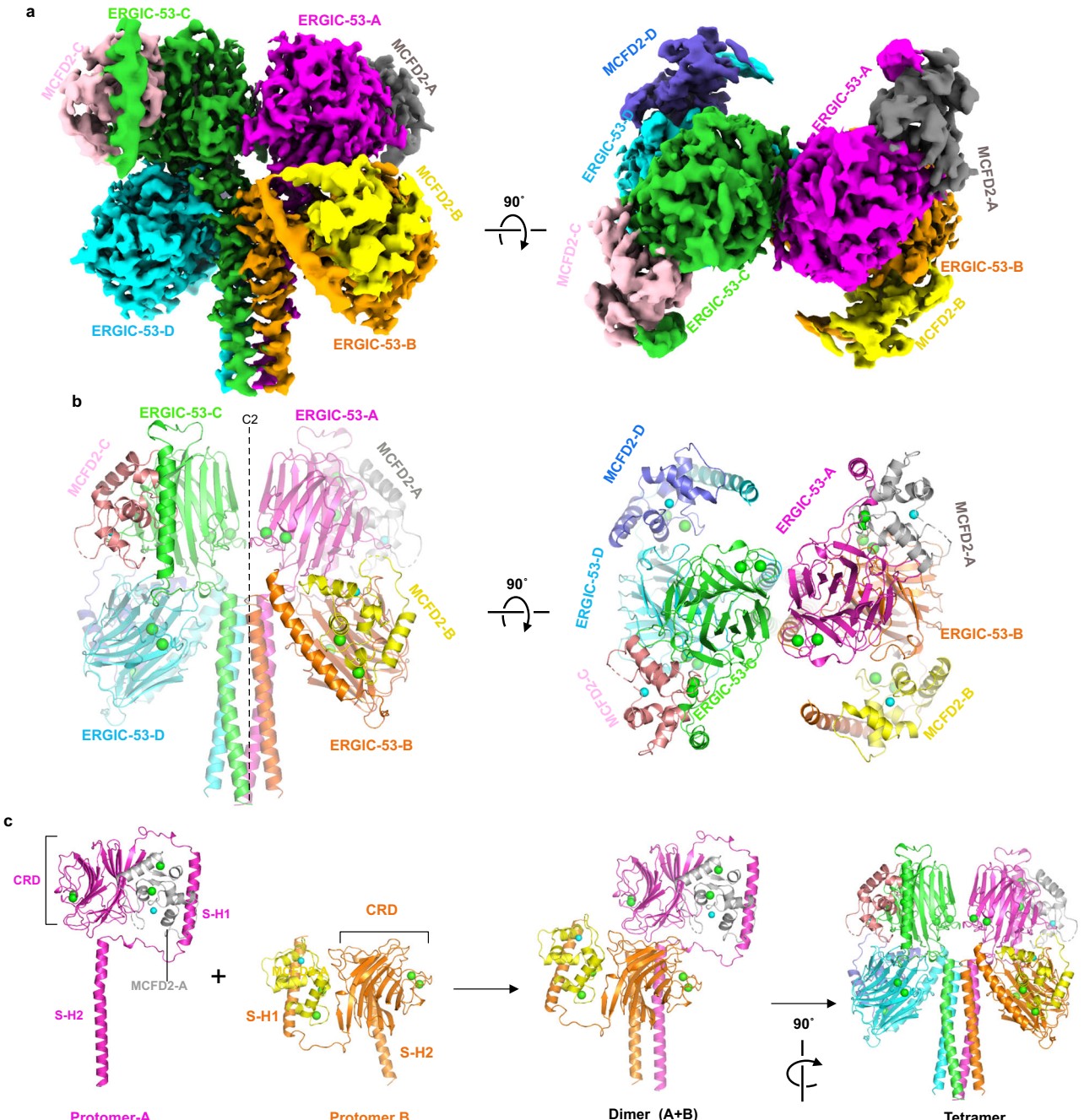

**Fig. 1 | Cryo-EM structures of the head region of ERGIC-53 in complex with MCFD2. a** Side and top views of the cryo-EM map of the head region of ERGIC-53 with MCFD2. Four ERGIC-53 protomers and four MCFD molecules are shown in magenta (ERGIC-53 protomer A), orange (protomer B), green (protomer C), cyan (protomer D), gray (MCFD2-A), yellow (MCFD2-B), pink (MCFD2-C), and slate blue (MCFD2-D), respectively. **b** Ribbon diagram of the structure of the tetrameric head region of the ERGIC-53. The four ERGIC-53 protomers and four MCFD2 molecules are shown in the same colors as in 1a. Green and cyan spheres represent calcium and zinc ions, respectively. **c** Details of the tetrameric head formation. The left two panels show structures of the head region of the ERGIC-53 protomers in the upper-CRD configuration (protomer A) and lower-CRD configuration (protomer B).

are located in the upper layer, whereas the other two lie in the lower layer (Fig. 1b). In the protomers with the upper-CRD (protomers A and C), the head region adopts a "Hisyaku (Japanese ladle)"-like shape, in which both CRD and S-H1 are positioned above the S-H2 (Fig. 1c). MCFD2 is accommodated in the pocket formed by CRD, S-H1 and two long loops between CRD and S-H1 and between S-H1 and S-H2. In the protomers with the lower CRD (protomers B and D), CRD and S-H1 are rotated by ~180° around the S-H1-S-H2 loop, and are located below the top of S-H2. The upper-CRD protomer vertically associates with the lower CRD protomer to form a dimer, and

then the two dimers are further assembled to form the four-leaf clover-like tetramer (Fig. 1b, c).

In the assembled head region, two different CRD-CRD interfaces are formed: a vertical interface (interface-V) between the vertically assembled CRDs (between protomers A and B and between protomers C and D) (Supplementary Fig. 6a) and the other interface (interface-H) between the horizontally assembled upper CRDs (between protomers A and C) (Supplementary Fig. 6b). Interface-V is formed by van der Waals contacts between residues in the β-strands and their flanking loops of each CRD (Supplementary Fig. 6a, inset 1). A similar interface is

also observed in the crystal packing of the CRD-MCFD2 complex (PDB code: 3WHU, 4YGE, etc.). The lower CRD (protomer B) also interacts with S-H2 from the counter protomer (protomer A) (Supplementary Fig. 6a, inset 2). On the other hand, interface-H is formed by only four hydrogen bonds, including that between Arg192 and Gln191' and two van der Waals contacts between the CRDs (Supplementary Fig. 6b).

As described above, two different forms (A and B) of the head region are determined, in which the positions and orientations of the CRDs relative to the S-H2 are different. The superimposition on the central S-H2 coiled-coil demonstrates that during the conversion from form A to form B, the upper and lower CRDs rotate by 4.8° as a rigid body (Supplementary Fig. 6c). Consequently, interface-H in form B is formed by interactions between different residues, such as van der Waals contact between Gln191 and Glu228' (Supplementary Fig. 6b right), while interface-V is almost maintained. These results suggest that interface-V involves tight binding, whereas interface-H is variable. As a result, the vertically assembled pair of CRDs can rotate to a certain extent (by 4.8°) as a rigid body.

## Overall architecture of full-length ERGIC-53 with MCFD2

Next, we endeavored to determine the whole structure of the full-length ERGIC-53-MCFD2 complex. Manual picking of full-length particles was first conducted based on the positions of the head and TM regions. An initial picking model was trained by Topaz[40] using particles selected from 2D classification, and the picking model was further improved by repeating Topaz picking, 2D classification, and Topaz training. Finally, Topaz successfully recognized the full-length particles and enabled us to perform a single particle analysis of full-length ERGIC-53 (Supplementary Fig. 7).

2D classification of the collected full-length particles generated clear 2D average images of their four-leaf clover-like structures consisting of the head, long stalk, and TM regions (Fig. 2a). Some particles adopt nearly straight conformations, while others possess stalks largely bent at various angles. These results suggest that the stalk region of ERGIC-53 is highly flexible.

By using straight particles, the cryo-EM structure of full-length ERGIC-53 in complex with MCFD2 was determined at 6.8 Å resolution (Fig. 2b and Supplementary Table 2). The maximum length of the full-length ERGIC-53 structure is ~340 Å, which is almost comparable to the $D_{max}$ value (347 Å) obtained by SAXS analysis (Supplementary Fig. 1f). Based on the obtained cryo-EM map, we built a structure model of full-length ERGIC-53 by using AlphaFold2 multimer[41] (Fig. 2c). Similar to S-H2, the stalk helices 3 (S-H3) and 4 (S-H4) also appear to form a four-helix coiled-coil. Thus, the three sets of long four-helix coiled-coils are vertically connected by relatively long loops to form the stable but flexible tetramer. Although the TM helices were not well resolved in the current cryo-EM map, they are also predicted to form a four-helix bundle and be vertically connected to S-H4. As a result, the full-length ERGIC-53 adopts a slender overall architecture. To our knowledge, the molecular height of full-length ERGIC-53 is taller than that of any other membrane protein of known structure, including the largest $Ca^{2+}$ channel ryanodine receptor RyRs, V-type ATPase, and spike protein from COVID-19 (Supplementary Fig. 7c).

To assess the structural contribution of MCFD2 to the overall architecture of ERGIC-53, we also attempted the cryo-EM analysis of full-length ERGIC-53 alone. While 2D average images showed features of the C-terminal portion including SH-4 and the TM domain, clear 2D images were not observed for the tetrameric head regions (Supplementary Fig. 8a). In the full-length particle analysis by using Topaz, the head regions were highly blurred after 2D averaging (Supplementary Fig. 8b). Consistently, our SEC-SAXS analysis suggests that the structure of full-length ERGIC-53 is more flexible in the absence of MCFD2 (Supplementary Fig. 1f, g). Thus, MCFD2 binding seems likely to stabilize the structure of the tetrameric head of ERGIC-53.

## Detailed molecular motions of full-length ERGIC-53

To gain insight into the molecular motions of full-length ERGIC-53, 3D variability analysis (3DVA) was performed in CryoSPARC[42], resulting in the identification of two different continuous motions of this cargo receptor (components I and II) (Fig. 2d, e and Supplementary Movie 1). 3DVA suggests that the entire structure of ERGIC-53 can be divided into three rigid-body segments, which correspond to the head region (segment 1), S-H3 (segment 2), and S-H4 (segment 3), respectively, followed by the C-terminal TM helix. The hinges between the three segments correspond to loops between S-H2 and S-H3 (hinge A) and between S-H3 and S-H4 (hinge B). In component I, segments 1 and 2 undergo a slight bending and stretching motion around hinge A. Segment 3 also undergoes a swing movement, rotating about hinge B in the opposite direction to segments 1 and 2 (Fig. 2d). The rotation directions of these three segments are nearly parallel to the interface-H between the upper CRDs (Fig. 2d bottom). The density of the TM domain was largely variable during the conversion from the first to last frames in this component, suggesting that the TM domain also swings around the loop flanked by S-H4 and the TM domain (i.e., the third hinge) (Fig. 2d). In component II, the segments 1–3 undergo similar bending and stretching movements through the two hinges, in which their rotation directions are perpendicular to interface-H (Fig. 2e, bottom). In addition, other conformations with even more greatly bent stalks were also observed in the 2D class-average images of the intact particles (Fig. 2a lower panels). These results suggest that full-length ERGIC-53 continuously adopts diverse conformations by bending stalk and TM helices at the three hinges.

## Molecular motion of the head region in full-length ERGIC-53

As mentioned above, the two different structures of the head region suggest its conformational variability (Supplementary Fig. 6c). To further assess the conformational variability of the head region, 3DVA was also performed by focusing exclusively on this region (Supplementary Fig. 9a and Supplementary Movie 2). As observed in forms A and B (Supplementary Fig. 6c), component I[head] shows a swing movement of the CRDs, in which the CRD dimer between upper and lower protomers shows a side-to-side rotation as a rigid body dimer (Supplementary Fig. 9a left and Movie 2 left). The rotation direction of the dimer is nearly parallel to interface-H (side-to-side rotation) (Supplementary Fig. 9a lower left). In component II[head], two CRD dimers rotate along a vertical axis (i.e., nearly parallel to the central coiled-coil) in opposite directions (referred to as twisting motion) (Supplementary Fig. 9a lower right and Supplementary Movie 2, right).

Based on a clustering analysis of the 3DVA of the head region, four different substate structures (A–D) were reconstructed at 3.29–3.51 Å resolution (Supplementary Fig. 9b, c). Superimposition of the central coiled-coil helices from the substrates A and B reveals that the upper and lower CRD dimers are rotated by 6.2° as a rigid body, which is a larger rotation between form A and form B described above, suggesting that form A and form B represent more averaged structures of the head region individually (Supplementary Figs. 9d vs. 4c). In the superposition of substate structures C and D, S-H1 and MCFD2 in the upper layers and those in the lower layers rotate by 2.5° and 1.9°, respectively, breaking the C2 symmetry of the head region (Supplementary Fig. 9e). These rotations seem to be ascribed to the flexible loops between the CRD and S-H1 and between S-H1 and S-H2. Thus, the two loops within the head region likely generate the plasticity of the head region, allowing the CRD-MCFD2 complex to fluctuate continuously in the tetrameric structure.

## Local plasticity of the head region independent of the global stalk bending

To investigate the linkage between the local plasticity of the head region and the global stalk bending, we constructed an ERGIC-53 ΔH34 mutant in which S-H3 and S-H4 within the stalk were deleted, and

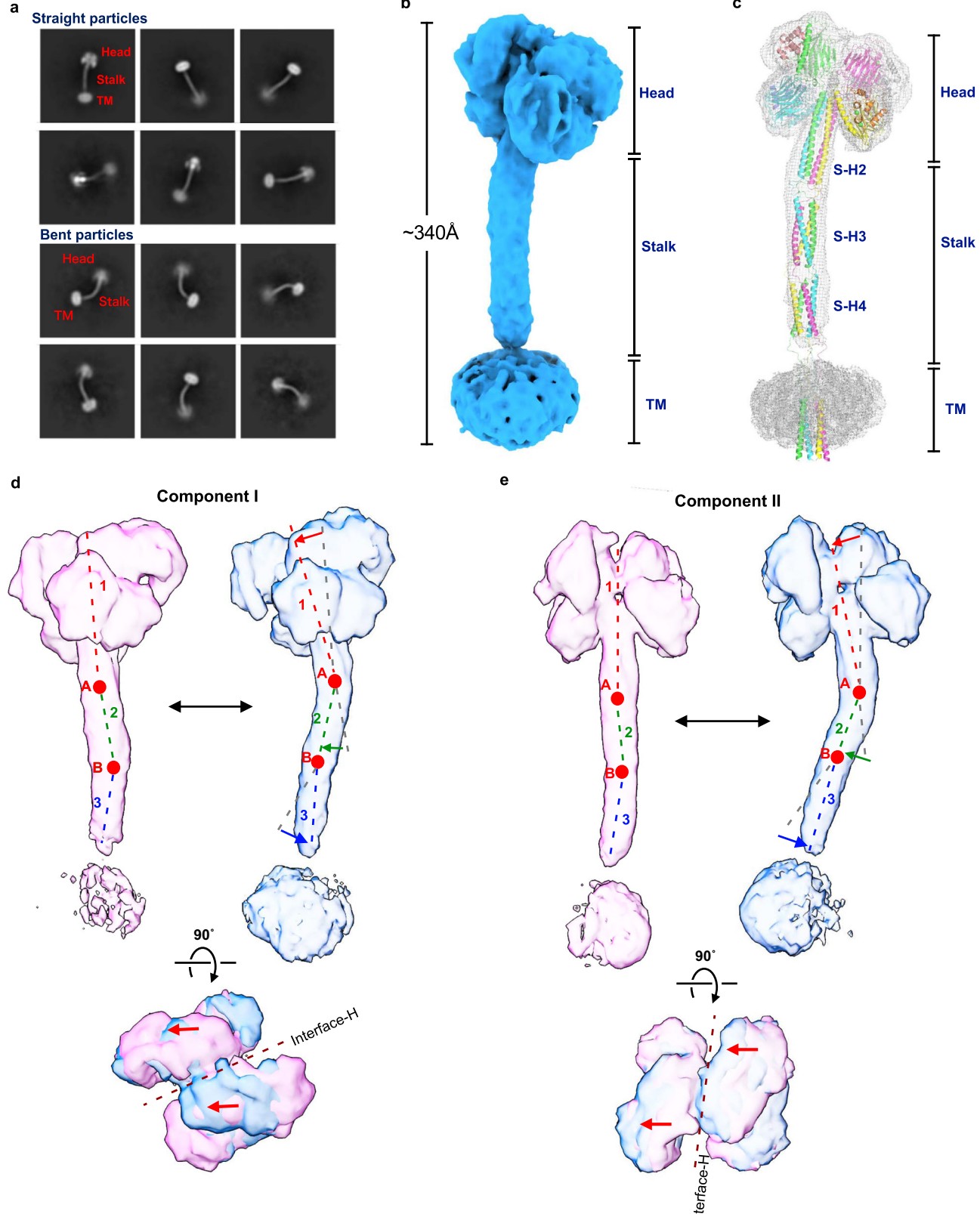

**Fig. 2 | Cryo-EM structure of full-length ERGIC-53 in complex with MCFD2.**
**a** Representative 2D class-average image of the full-length particles. The upper and lower panels represent straight and bent particles, respectively. **b** Cryo-EM map of full-length ERGIC-53 in complex with MCFD2. **c** The overall architecture of full-length ERGIC-53 overlaid in the EM map shown in (**b**). **d**, **e** Results of 3DVA of the full-length ERGIC-53 with two variability components I (**d**) and II (**e**). The 3D EM maps of the first (pink) and last frames (blue) of continuous conformational changes generated by 3DVA are displayed. Broken lines represent the center of three segments. Red circles represent hinges between the segments 1, 2, and 3. The lower panels show the top views of the two maps superimposed on each other. The brown broken line represents the interface-H between the upper CRDs.

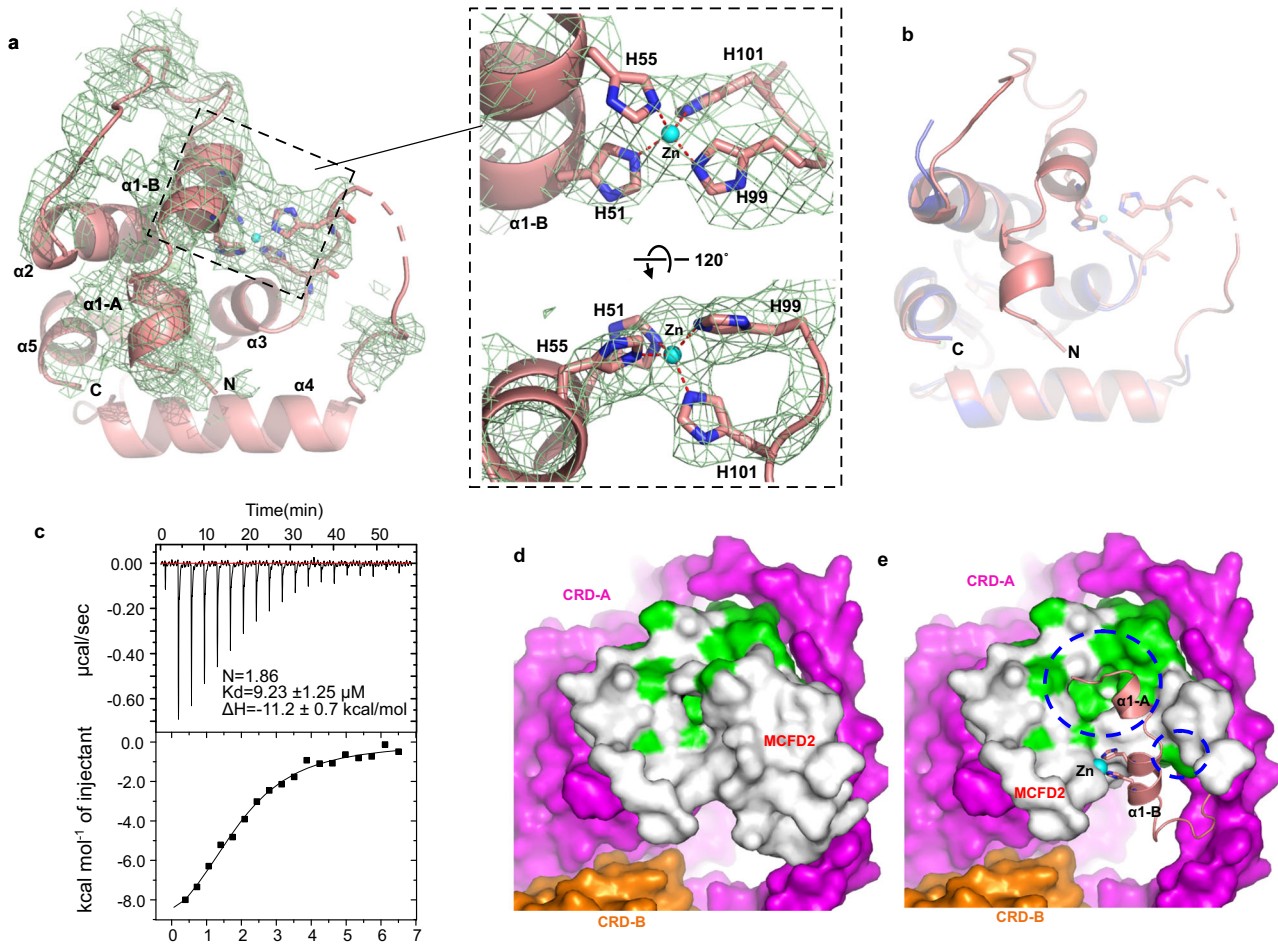

**Fig. 3 | Visualized N-terminal segment of MCFD2 contains a Zn²⁺ binding site.** **a** The overall structure of MCFD2 bound to full-length ERGIC-53. The EM density map around the region that is missing in the previous crystal structures is shown in a light green mesh. The inset shows a close-up view of the His cluster, showing that Zn²⁺ is probably coordinated by four His residues. **b** Superposition of the Zn²⁺-bound MCFD2 structure revealed by the present cryo-EM analysis (pink) with the previously reported structure of Zn²⁺-free MCFD2 (blue, PDB ID: 4YGE). **c** ITC raw data (upper) and binding isotherm data (lower) for titration of ZnCl₂ (500 μM) into the EDTA-treated MCFD2 sample. Errors represent fitting residuals (standard deviation). **d**, **e** Comparison of FV/FVIII binding sites in MCFD2 with (**d**) or without (**e**) the N-terminal lid. MCFD2 (white) and two CRDs (magenta and orange) are shown in the surface representation. Previously identified FV/FVIII binding sites in MCFD2 are colored in green. Blue dashed circles indicate the regions of the FV/FVIII binding site masked by the N-terminal lid.

determined its cryo-EM structure at 3.78 Å resolution (Supplementary Fig. 10a, b). The ΔH34 mutant retains a tetrameric structure with the assembled head region, similar to wild-type. Additionally, the TM helices of the mutant were not well resolved in the density map, as was the case for the wild-type. 3DVA of this deletion mutant revealed its three different motions (Supplementary Fig. 10c and Supplementary Movie 3). In the component I^ΔH34, the TM domain shows a large swing movement, while the head region remains static (Supplementary Fig. 10c, left and Movie 3 left). On the other hand, components II^ΔH34 and III^ΔH34 show the side-to-side rotation and twisting motion of the CRD dimers, respectively (Supplementary Fig. 10c, center and right and Supplementary Movie 3, center and right), as observed in the 3DVA of the head region of full-length ERGIC-53. The TM domain shows only marginal rotation in either component II^ΔH34 or III^ΔH34. Thus, the CRD dimers in the head region appear to move separately from the region below the S-H2, suggesting that there is no linkage between the head-domain plasticity and the stalk-domain bending.

## Visualization of an N-terminal segment with a zinc-binding site in MCFD2

In the previously reported crystal structures of MCFD2 alone and its complex with CRD[20,34–37], the N-terminal region of MCFD2 (residues 27–66) and a loop between α3 and α4 helices (residues 100–111) were disordered and hence invisible (Supplementary Fig. 11a). However, the present cryo-EM map displays significant density for the regions preceding the α2 helix (Fig. 3a). In this context, AlphaFold2 prediction[41] suggests that the disordered N-terminal region contains two short α-helices (α1-A and α1-B) (Supplementary Fig. 11b). Although we had difficulty in de novo modeling of the MCFD2 N-terminal region from the present map, the predicted N-terminal α1-A, and α1-B short helices and a loop between the α1-B and α2 helices are nicely fitted onto the observed density map (Fig. 3a). Thus, the present cryo-EM structure of MCFD2 in complex with full-length ERGIC-53 is nearly identical to the original structures (RMSD of 0.53 Å) except the N-terminal segment (Fig. 3b). The difference in the maps of MCFD2 obtained by the crystal and cryo-EM analyses is likely due to the presence of S-H1 of ERGIC-53 in the cryo-EM sample. Indeed, this helical segment interacts with the N-terminal region of MCFD2, seemingly restricting its conformational flexibility (Fig. 2c and Supplementary Fig. 6a).

In the updated MCFD2 structure, four conserved His residues (His51, His55, His99, and His101) are clustered at the α1-B helix and a loop between the α3 and α4 helices (Fig. 3a and Supplementary Fig. 11d). Notably, the geometry of the histidine cluster and the presence of density at the center of the cluster are consistent with

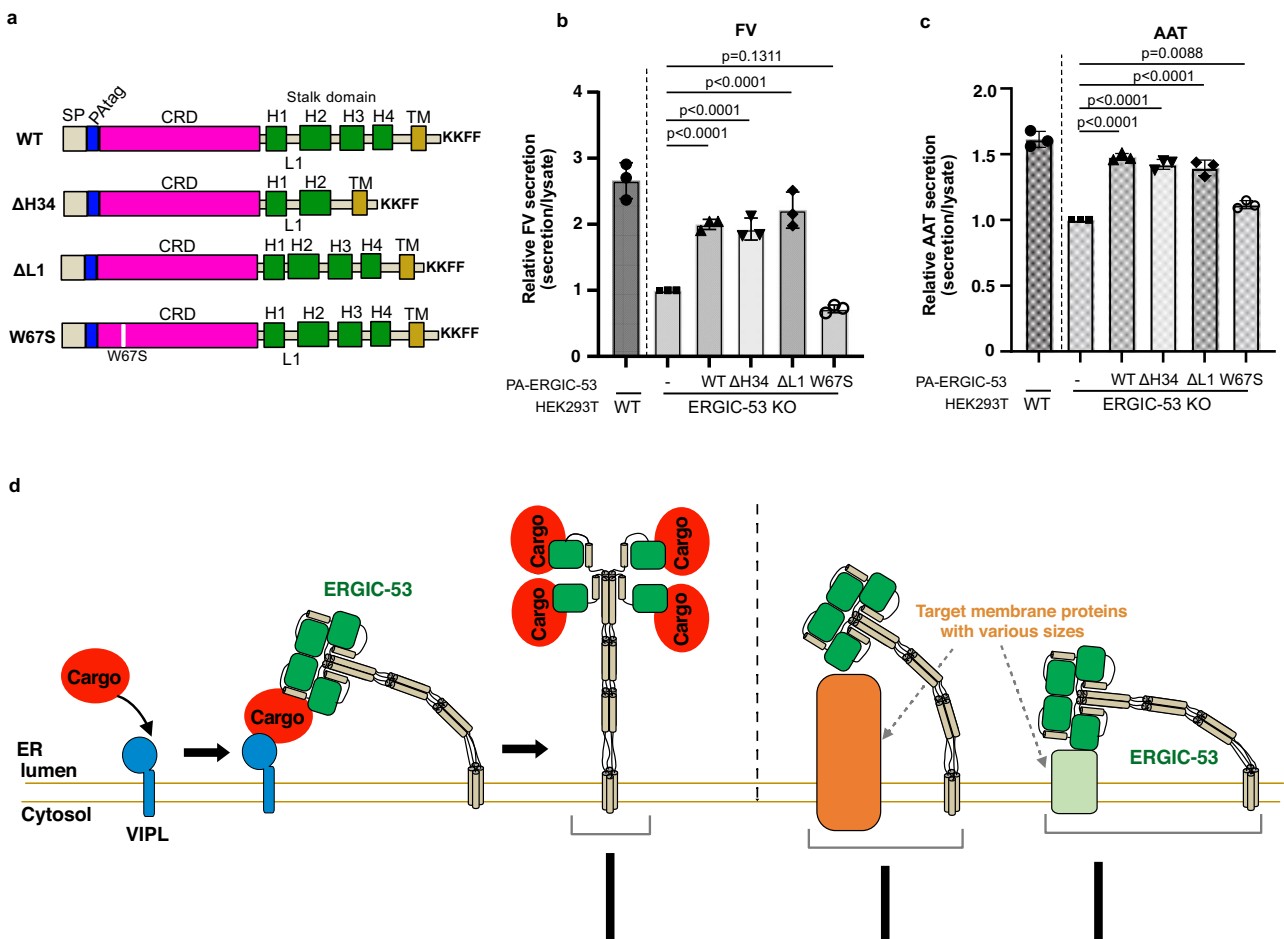

**Fig. 4 | Role of the long stalk and the head assembly in cargo transport.**
**a** Schematic representation of the ERGIC-53 mutants used for the rescue experiments. **b, c** Rescue experiments of FV-HiBiT (**b**) and AAT-HiBiT (**c**) secretion by coexpression of ERGIC-53 or its mutants in ERGIC-53 KO cells. The amounts of secreted FV/AAT relative to those of intracellular FV/AAT were quantified with a commercially available HiBiT-tagged protein detection reagent and normalized to those from the KO cells transfected with an empty vector. Each symbol represents the individual data points. Error represents the standard deviation of the mean ($N = 3$ biological replicates). Statistical significance was measured with one-way ANOVA followed by Dunnett's test. Exact P values are indicated in the graphs for each construct. **d** A working model of cargo capture by ERGIC-53.

coordination of a metal, most likely a zinc ion (Fig. 3a inset). In support of this, a colorimetric assay with 4-(2-pyridylazo) resorcinol (PAR), a chromophoric chelator for divalent metal ions, revealed that purified MCFD2 contained significant amounts of divalent metal ions such as $Zn^{2+}$ (Supplementary Fig. 11c). Consistently, our isothermal titration calorimetry (ITC) analysis revealed that an EDTA-treated MCFD2 sample can bind up to two molar equivalents of $Zn^{2+}$ with micromolar affinity (Fig. 3c). Thus, MCFD2 has been shown to be a zinc-binding protein with the conserved His cluster.

A previous NMR study revealed that the EF-hand helices of MCFD2 are involved in the interaction with FV/FVIII[43]. In the present cryo-EM structures, the reported FV/FVIII binding site in MCFD2 is not fully exposed to the solvent but rather shielded by its N-terminal short helices (Fig. 3d, e). The N-terminal segment of MCFD2 appears to be hooked by the $Zn^{2+}$-bound His cluster formed between the α1-B helix and α3-α4 loop (Fig. 3a). These structural features raise a possibility that $Zn^{2+}$ binding to the His cluster may prevent the cargo binding to the head of ERGIC-53.

To explore how $Zn^{2+}$ affects cargo transport, we performed an FV secretion assay using WT and ERGIC-53 KO 293T cells with or without $Zn^{2+}$ supplementation. Intracellular $Zn^{2+}$ concentration is known to be affected by extracellular $Zn^{2+}$ levels in culture medium[44]. When 10 μM

$ZnCl_2$ was added to the serum-free Expi293 medium (Supplementary Fig. 11e), FV secretion from WT cells was slightly but significantly prevented. In contrast, the ERGIC-53 KO cells hardly displayed the $Zn^{2+}$-induced prevention of FV secretion. Altogether, these results suggest that the cargo transport mediated by ERGIC-53/MCFD2 can be inhibited by $Zn^{2+}$, consistent with the present observation that the cargo-binding site of ERGIC-53 is masked by the N-terminal segment of "$Zn^{2+}$-bound" MCFD2.

**Physiological significance of the long stalk and head assembly**
Our cryo-EM structures reveal that full-length ERGIC-53 adopts the four-leaf clover-like structure with an unusually long and flexible stalk. To examine the physiological roles of the long stalk in the cargo transport, rescue experiments were performed by monitoring FV and alpha-anti-trypsin (AAT) secretion under coexpression of the cargo proteins and ERGIC-53 WT or its stalk-deletion mutant in the ERGIC-53 KO cells (Fig. 4a and Supplementary Fig. 12a). Upon cotransfection, the expression levels of WT and its mutant were comparable to that of endogenous ERGIC-53 in HEK293T cells (Supplementary Fig. 13a, d). As expected, the coexpression of ERGIC-53 WT enhanced the secretion of FV and AAT, compared to the empty vector (Fig. 4b, c and Supplementary Fig. 13b−e). Similarly, expression of the ΔH34 mutant with a

much shorter stalk (Fig. 4a) rescued the FV section at a level comparable to WT (Fig. 4b). A similar result was obtained for the AAT secretion (Fig. 4c). Thus, deletions of the S-H3 and S-H4 only marginally affected the cargo transport ability of ERGIC-53. To verify the validity of our ERGIC-53-mediated cargo secretion assay, we also investigated the FV and AAT secretion from the ERGIC-53 KO cells expressing ERGIC-53 W67S, a missense mutant deficient in both MCFD2 and mannose bindings[45]. As reported previously[45,46], this negative control mutant failed to rescue the FV secretion while the AAT secretion was partially rescued by this mutant (Fig. 4b, c), probably because ERGIC-53 captures AAT in an MCFD2-independent manner[47]. Collectively, it can be interpreted that the shorter version of ERGIC-53 (ΔH34) still retains the ability to enhance the secretion of FV and AAT, although the distance between the head and TM domain and the flexibility of the stalk are substantially reduced compared to those of wild-type.

Additionally, to examine the physiological significance of the head region assembly for cargo transport, we constructed a mutant lacking the L1 loop between S-H1 and S-H2 (residue Gln313 to Glu322) (ΔL1 mutant) (Fig. 4a and Supplementary Fig. 12b, c). In the head region, two S-H1s of protomers A and B are located ~65 Å apart by the L1 loops to form the vertically assembled CRD dimers. (Supplementary Fig. 12b). Hence, the deletion of the L1 loop is predicted to disrupt the interface-V between the upper and lower CRDs, as shown in the AlphaFold2 (AF2) predicted model (Supplementary Fig. 12c, d). The ΔL1 mutant displayed a similar migration pattern to the WT on a nonreducing SDS gel (Supplementary Fig. 13a lane 5), suggesting that this mutant also exists as the mixture of covalent and noncovalent tetramers, likely formed through the central four-helix coiled-coils. Coexpression of the ΔL1 mutant also rescued the FV and AAT secretion at a level similar to that of the WT (Fig. 4b, c). These results suggest that the head region assembly is not an essential factor for the efficient transport of soluble cargo proteins by ERGIC-53 and MCFD2.

## Discussion

The present cryo-EM analysis reveals the tetrameric slender architecture of full-length ERGIC-53 in complex with MCFD2. The long stalk of ERGIC-53 is composed of three sets of four-helix coiled-coils that are vertically aligned. Full-length ERGIC-53 can adopt various conformations, including straight and largely bent conformations by bending the stalk domain. The CRDs within the head region also fluctuate independently of the stalk bending. These flexible motions of ERGIC-53 are most likely generated by the loops that function as hinges between the CRD, stalk helices, and TM domain.

The tetramer formation of ERGIC-53 revealed in this study is similarly observed with its yeast homolog Emp46p and Emp47p[48,49], suggesting that the tetrameric state may be widely conserved among the eukaryotic ERGIC-53 homologs. The heterotetramer complex of Emp46p and Emp47p was stable at pH 6–7, as observed for ERGIC-53 (Supplementary Fig. 2). However, the heterotetramer composed of Emp46p and Emp47p was reported to be dissociable at more acidic pH. In this transition, a glutamate residue Glu306 of Emp46p, which is located inside the hydrophobic core of its coiled-coils, is suggested to function as a pH sensor that regulates its oligomeric state. By contrast, ERGIC-53 contains no charged residues inside the hydrophobic core of the coiled-coils. Notably, the three stalk helices of ERGIC-53 (~120 residues) are much longer than those of Emp46p/Emp47p (~60 residues), suggesting tighter interactions between the coiled-coils in ERGIC-53. Furthermore, the ERGIC-53 tetramer is stabilized by intermolecular disulfide bonds. Thus, the tetrameric state of ERGIC-53 is likely robust under the environmental conditions of the early secretory pathway and retained during the cycling between the ER and GA.

In the prostate and several tissues, an ERGIC-53 homolog, ERGL (ERGIC-53- like, LMAN1L), is highly expressed[50]. A recent reported that ERGL is also involved in the secretion of the FVIII section in HCT116 cells. However, ERGL is not expressed at a detectable level in

HEK293 cells[51], suggesting that the functional contribution of endogenous ERGL may be negligible in our rescue experiments. The putative stalk domain of ERGL, which shows a low sequence similarity to that of ERGIC-53, is also predicted to form three sets of coiled-coils (Supplementary Fig. 14a). Moreover, AF2 multimer predicted a possible heterotetramer structure of ERGIC-53 and ERGL via three sets of four-helix coiled-coils, despite the low sequence similarity between these two (Supplementary Fig. 14b). Thus, ERGL may exit as a homotetramer or form a heterotetramer with ERGIC-53 to function as an additional or auxiliary cargo receptor in some tissues and cells.

ERGIC-53 has been proposed to cooperate with other luminal L-type lectins, such as VIP36(LMAN2), VIPL(LMAN2L), and ERGL (LMAN1L), to maintain the homeostasis of secretory cargo proteins[26,50,52]. In the proposed model, properly folded cargo proteins with M9 glycans are first recognized by VIPL, and then transferred to ERGIC-53. VIPL consists of a CRD and a single TM but lacks a stalk region. ERGIC-53 presumably receives cargo proteins from the small VIPL by utilizing the largely bent long stalk (Fig. 4d left). In this context, it is reasonable that the ΔH34 mutant with the much shorter stalk still retains the cargo transport activity at a similar level to WT. Our rescue experiments also revealed that the head assembly is not essential for efficient cargo transport. Given the docking model of dimannose bound to the CRD based on the previous crystal structure (4YGE), the 1-OH of the Man (4) moiety of $Man_9(GlcNAc)_2$ glycan is located near the L1 loop between S-H1 and S-H2 (Supplementary Fig. 11f). In the tetrameric head, the L1 loop may interfere with the accommodation of N-glycans of cargo proteins. Thus, the present CRD assembled structure possibly represents a post-cargo release state of the ERGIC-53-MCFD2 complex. Cargo binding may alter the relative positions of the CRDs to capture a target cargo protein in concert with MCFD2.

In addition to soluble secretory proteins, ERGIC-53 is known to facilitate the transport of certain specific membrane proteins, including neuroreceptors such as $GABA_AR$ in neuronal cells[22] and several Golgi resident membrane proteins such as GOLGA5, NDST1, and FKRP[23]. ERGIC-53 is also associated with surface glycoproteins of arenavirus, hantavirus, coronavirus, and hepatitis B virus[24,25]. Given that these target membrane proteins vary in their size and structures, the conformational flexibility of the stalk domain and the local rotation of the CRDs appear to help adjust the height and position of the head region of this cargo receptor to those membrane proteins (Fig. 4d right). Notably, although the CRD of ERGIC-53 is an essential element for the transport of these membrane proteins, its glycan binding ability is dispensable for the interaction with them[24,25]. Collectively, ERGIC-53 is likely to recognize and bind target membrane proteins differently from soluble proteins.

Here, MCFD2 has been found to be a zinc-binding protein. The ITC analysis for $Zn^{2+}$ binding to MCFD2 and purification of MCFD2 in a metal ion ($Zn^{2+}$)-bound form from E. coli cells suggest that the $Zn^{2+}$ binding ability of MCFD2 does not rely on the complex formation with ERGIC-53. While the overall structure of MCFD2 is not largely altered upon $Zn^{2+}$ binding, its cargo-binding site is masked by its N-terminal helix in the present cryo-EM structure, and this local structure is probably stabilized by the $Zn^{2+}$-bound His cluster (Fig. 3a). The relatively low local resolution of the N-terminal region of MCFD2 may suggest that this segment is readily moved away upon cargo binding. Thus, the N-terminal helix of MCFD2 appears to function as a lid to regulate the cargo binding/release in a $Zn^{2+}$-dependent manner (Supplementary Fig. 15). It has been presumed that ERGIC-53 releases cargo proteins in the GA with lower $Ca^{2+}$ concentration and lower pH[11,18,19,26,27]. In contrast to $Ca^{2+}$, the labile $Zn^{2+}$ concentration has been reported to be much higher in the GA than in the ER and other organelles[53–55]. Accordingly, these findings suggest that $Zn^{2+}$ binds transiently to MCFD2 in the GA and thereby promotes the cargo release by closing its N-terminal lid. The proposed working hypothesis of $Zn^{2+}$-dependent regulation of the ERGIC-53 MCFD2 system is relevant to our recent

finding that $Zn^{2+}$ is vital to protein quality control mediated by ERp44 in the early secretory pathway (ESP)[54,56]. In sharp contrast to the ERGIC-53-MCFD2 system, $Zn^{2+}$ enhances the complex formation between ERp44 and its client proteins in the GA, whereas ERp44 releases its clients in the ER at much lower $Zn^{2+}$ concentrations[54,56]. Future studies will further elucidate the physiological roles of $Ca^{2+}$, $Zn^{2+}$, and other metal ions in protein homeostasis mediated by various molecular chaperones and cargo receptors in the ESP.

As an additional note, the revealed slender structure of full-length ERGIC-53 seems highly relevant to C-type lectins, which are also composed of a CRD, long stalk, and TM domain. Many of the C-type lectin family proteins are localized on the cell surface and function as receptors for a broad range of ligands, including pathogens[57,58]. Intriguingly, a recent study reported that in dendritic cells and their related cells, such as airway epithelial cells, a fraction of ERGIC-53 is located on the cell surface and functions as a receptor for house dust mite (HDR) allergens[59]. The molecular height of ERGIC-53, which is comparable to those of the C-type lectins, is likely advantageous for this receptor to capture target HDR and other potential ligands, without substantial interference from surrounding C-type lectins on the cell surfaces. It is still difficult to visualize cell surface receptors such as full-length C-type lectins due to their unusually long stalks and large flexibility. Nonetheless, as shown here, cryo-EM single particle analysis can visualize the overall structures of the flexible receptor not only in the 2D class-average images but also in 3D reconstruction at a middle resolution that enables the reliable identification of each domain. Flexibility analysis of the EM data, such as 3DVA, can visualize their global motion. Further structural analysis of long flexible receptors will be feasible and provide a deep understanding of their structures and mechanisms of action.

## Methods

### Cell cultures and plasmids

The cDNA of human ERGIC-53 was subcloned from a cDNA library from HeLa cells and inserted into the pcDNA3.1 vector (Invitrogen). Subsequently, the ERIGIC-53 gene with a C-terminal PA tag was inserted into a PiggyBac-Cumate Switch Inducible vector (System Bioscience). The pcDNA-SP-PA-ERGIC-53 plasmid was generated using the In-Fusion cloning kit (Takara) with PCR fragments from the pcDNA3.1-ERGIC-53-PA plasmid. Mutants of ERGIC-53 were constructed by PCR-based site-directed mutagenesis. A PCR fragment of the ΔH34 mutant with a C-terminal FLAG was also inserted into a pEMmulti-puro vector (Fijifilm-Wako). Sequences of the primers used in this study are summarized in Supplementary Table 3.

The cDNA of human MCFD2 (IRAK026G06) was provided by the RIKEN BRC through the National BioResource Project of the MEXT/AMED, Japan. The lumen domain of MCFD2 (residues 27–146) was subcloned into the pET15b vector with an N-terminal histidine tag. The cDNA of human Factor V (FV) (FXC01788) was purchased from Kazusa DNA Research Institute. The secreted region of FV (residues 29-2224) was inserted into the phlsec-vector (Addgene, 72348) with the C-terminal HiBiT-Tag (Promega), using the In-Fusion cloning kit. A codon-optimized cDNA fragment of the secreted region of human AAT was synthesized (Eurofins) and inserted into the phlsec-vector with the C-terminal HiBiT-Tag.

Human embryonic kidney (HEK) 293T cells were purchased from ATCC (American Type Culture Collection, CRL-3216). Human ERGIC-53 (LMAN1) KO 293T cell lines (ab266248), as well as wild-type cells (ab255449) were purchased from Abcam.

### Protein expression and purification

The PiggyBac-cumate-ERGIC-53-PA tag plasmid, along with the Super PiggyBac Transposase Expression Vector (System Bioscience), was transfected to generate a stable HEK293T cell line with cumate-inducible ERGIC-53 expression. The established stable cells were cultured in Dulbecco's modified Eagle's medium (DMEM, Nacalai Tesque) supplemented with 4–5% fetal calf serum (FCS; Thermo Fisher Scientific, or Nichirei Biosciences Inc) and 1% penicillin-streptomycin mixed solution (Nacalai Tesque) under 5% $CO_2$ at 37 ˚C. When cells were grown to 70–80% confluency in 15 cm dishes, the medium was replaced with serum-free DMEM supplemented with 6 mM sodium butyrate (Nacalai Tesque). After 6 h incubation at 37 ˚C, expression of ERGIC-53 was induced by the addition of cumate (240 μg/ml) and phorbol 12-myristate 13-acetate (PMA, 50 nM), and then cells were cultured at 30 ˚C in 5% $CO_2$ for another 2 days before harvesting. The cells were washed with PBS and stored at −80 ˚C. The pEB-multi-puro-ERGIC-53 ΔH34-FLAG was transfected to generate a stable HEK293T cell line for expression of the ERGIC-53 ΔH34-FLAG mutant. The established stable cells were cultured in DMEM supplemented with 4–5% FCS and 1% penicillin-streptomycin mixed solution (Nacalai Tesque) under 5% $CO_2$ at 37 ˚C. When cells were grown to 70–80% confluency in 15 cm dishes, sodium butyrate (6 mM) and PMA (50 nM) were added, and then cells were cultured at 30 ˚C in 5% $CO_2$ for another 2 days before harvesting. The cells were washed with PBS and stored at −80 ˚C.

All purification procedures were conducted at 4 ˚C or on ice. Cell pellets expressing ERGIC-53-PA from 3 L of medium or those expressing the ERGIC-53 ΔH34 mutant from 1 L of medium were resuspended in buffer A (20 mM Tris-HCl pH 7.5, 150 mM NaCl, 5 mM $CaCl_2$, 10% glycerol) supplemented with 5 mM $MgCl_2$ and 1x protein inhibitor cocktail (Nacalai Tesque) and ~50 μg/ml DNase I (Fujifilm-Wako). The cells were broken by brief sonication and were solubilized by the addition of 1% (v/v) n-dodecyl-β-ᴅ-maltoside (DDM; Nacalai Tesque) for 2 h. The solubilized fraction was clarified by ultracentrifugation (Hitachi CS100FNX, 150,000×$g$, 20 min) and then mixed with 7.5 ml anti-PA-tag antibody beads (Fujifilm-Wako) for 2 h (for ERGIC-53-PA) or 1 ml Anti-DYDDDK tag antibody beads (MBL) for 2 h (for the ERGIC-53 ΔH34 mutant) The beads were transferred into a gravity flow column (Econo-column, Bio-Rad) and washed with 20 CV of buffer A supplemented with 0.02% (w/v) GDN (Anatrace). The bound proteins of ERGIC-53 WT were eluted with overnight incubation of 2 CV elution buffer (buffer A supplemented with 0.02% GDN, and 0.2 mg/ml PA peptide), followed by repeating 5 min incubation with 1 CV elution buffer. For the ΔH34 mutant, the bound proteins were eluted by repeating the 5 min incubation of 1 CV elution buffer (buffer A supplemented with 0.02% GDN and 0.1 mg/ml DYKDDDDK peptide). The eluted protein was concentrated using Amicon Ultra-15 centrifugal filters (100 kDa cut-off, Merck-Millipore). After treatment with 1 mM diamide for 15 min to completely oxidize the protein, the concentrated protein was applied onto a Superose 6 Increase 10/300 GL column (GE Healthcare) equilibrated with SEC buffer (20 mM Tris-HCl pH 7.5, 150 mM NaCl, 10 mM $CaCl_2$ and 0.02% GDN). For the preparation of the ERGIC-53-MCFD2 complex, the ERGIC-53 sample was mixed with excess purified MCFD2 (described below) for 10 min, followed by diamide treatment for 15 min, and then the complex sample was injected into the same column. The peak fractions were concentrated using Amicon-Ultra-4 and -0.5 centrifugal filters (100 kDa cut-off) for cryo-EM samples.

The pET15b plasmid of MCFD2 was introduced into E-cos Escherichia coli BL21 (DE3) (Nippon gene). Gene expression was induced with 0.1 mM isopropyl β-ᴅ-1-thiogalactopyranoside (Fujifilm-Wako) at an OD600 of ~0.5, with further cultivation for 24 h at 293 K before harvesting. Cells were washed with 50 mM Tris-HCl pH 8.5 and then stored at −80 ˚C. The cell pellet was resuspended in buffer A (50 mM Tris-HCl pH 8.0, 300 mM NaCl, 5 mM $CaCl_2$) supplemented with a protease inhibitor cocktail (Nacalai Tesque) and was sonicated on ice for 5 min. After the removal of cell debris by centrifugation (17000×$g$ × 30 min at 4 ˚C), the supernatant was applied to a Ni-NTA resin (Qiagen). After 10 CV washes with buffer A supplemented with 20 mM imidazole, the bound protein was eluted with buffer A supplemented with 200 mM imidazole. The eluted protein was concentrated and using Amicon Ultra-15 centrifugal filters (3 kDa cut-off,

Merck-Millipore), and then buffer was exchanged to buffer B (50 mM Tris-HCl pH 8.0. 5 mM CaCl$_2$). The protein was loaded onto an anion-exchange column (MonoQ 10/100 GL, GE Healthcare) equilibrated with buffer B, and eluted with a 0.1–0.4 M linear gradient of NaCl. The eluted protein was concentrated and treated with thrombin (Nacalai Tesque) at 4 °C overnight to cleave the His tag. The sample was further purified with a size-exclusion column equilibrated with 20 mM Tris-HCl, 150 mM and 5 mM CaCl$_2$. The eluted protein was concentrated and stored at −80 °C.

## SEC-MALS/SAXS

Size-exclusion chromatography coupled with multiangle light scattering (SEC-MALS) analysis was performed by using a high-performance liquid chromatography (HPLC) system, Alliance 2695 (Waters) with DAWN HELEOS II (Wyatt Technology) and 2414 Refractive Index (RI) detector and 2489 UV/Visible detector (Waters). Samples (1.6–1.7 mg/ml × 10 μl) were loaded on a Superdex 200 increase 3.2/300 column (Cytiva) equilibrated with SEC buffer (20 mM Tris-HCl pH 7.5, 150 mM NaCl, 10 mM CaCl$_2$, and 0.02% GDN) under a flow rate at 0.1 ml/min. The conjugate analysis of the molecular masses by combining MALS-UV-RI signals was carried out by ASTRA 6.1 (Wyatt Technology) with a dn/dc value of 0.185 mL/g for protein and 0.17 mL/g for GDN, and an extinction coefficient (1 mg/ml) of 0.766 for protein and 0.01014 for GDN. Although the dn/dc value of GDN was arbitrarily used based on the other detergents, this value did not affect the calculation of the molecular masses of the protein.

Size-exclusion chromatography combined with small-angle X-ray scattering (SEC-SAXS) was performed on the BL-15A2 beamline at the photon factory, KEK (Tsukuba, Japan)[60]. The sample (1.6–1.7 mg/ml × 50 μl) was loaded on a Superdex 200 increase 3.2/300 column equilibrated with the SEC buffer using an HPLC system Nexera-i (SHIMADZU) under the flow rate at 0.03 ml/min. A fiber spectrometer, QEpro (Ocean Insight), mounted at an angle of 45° to the sample cell, was also utilized to obtain the concentration for each frame. The sample-to-detector distance was set to 2650 mm, and the measured X-ray wavelength was adjusted to 1.0 Å. The 2D scattering intensity images were recorded using PILATUS3 2 M (DECTRIS) with an exposure time of 3 s. UV-visible absorption spectra were also recorded at 3-s intervals with a 1-s integration time. All scattering images were azimuthally averaged to convert the one-dimensional scattering intensity data, and then the subtraction of the background profile was also calculated. The absolute intensity calibration was performed using water as a standard. These processes were carried out using SAngler[61]. Scattering profiles above half of the elution peaks were averaged by using MOLASS[62] The radius of gyration ($R_g$) and and the maximum molecular dimension ($D_{max}$) from the Guinier approximation, and the pair distribution function ($P$(r)) were calculated by AUTORG and PRIMUSqt from ATSAS, respectively (Supplementary Fig. 1h)[63].

## Colorimetric analysis

To detect the metal ions contained in the purified sample of MCFD2, PAR analysis was conducted[64]. The purified sample of MCFD2 (50 μM) was denatured with 4 M guanidine-HCl at pH 7.6 for 5 min at 37 °C. The denatured sample was mixed with a freshly prepared solution of (4-(2-pyridylazo) resorcinol (PAR) (Tokyo Chemical industry) (final 100 μM), and then the UV-vis spectrum was immediately measured with a Hitachi UV3900. Note: Due to the relatively weak affinity, the amount of the bound metal ions (probably $Zn^{2+}$ or $Ni^{2+}$) in the purified MCFD2 samples varied between each purification.

## Isothermal titration calorimetry ITC

Isothermal titration calorimetry (ITC) experiments were carried out using the MicroCal™ iTC200 calorimeter (Malvern) at 293 K. After a 0.4 μL initial injection, 2 μL of ZnCl$_2$ solution (500 μM) was injected into the EDTA-treated MCFD2 solution (15 μM) in 20 mM Tris-HCl pH 7.5,

150 mM NaCl, and 0.2 mM CaCl$_2$, at 180 s intervals with stirring 750 rpm. The data analysis was performed with Microcal Origin Software (version 7.0) using a one-site binding model. The experiments were repeated at least twice with similar results.

## Cryo-EM sample preparation and data collection

The purified samples of full-length ERGIC-53 with or without MCFD2 (4–5 mg/ml) were mixed with 3 mM Fluorinated Fos-Choline-8 (Anatrase), and then 3 μl aliquot of the sample was applied to glow-discharged QuantiFoil R1.2/1.3 Au 300 mesh grids. The grids were blotted for 4 s and immersed in liquid ethane using Vitrobot Mark IV systems (FEI/Thermo Fisher) operated at 4 °C and 100% humidity. The grids were initially screened using Talos Arctica (FEI) with a K2 direct electron detector (Gatan). The best grids were loaded to a Titan Krios (FEI) microscope operated at 300 kV and equipped with a Gatan Quantum-LS Energy Filter (GIF) and a Gatan K3 BioQuantum direct electron detector. Movies were automatically collected using SerialEM software[65].

For the sample of the ERGIC-53 ΔH34-MCFD2 complex (1.7 mg/ml), two rounds of 3 μl aliquot loading and manual blotting were performed, and then a 3 μl aliquot of the sample was applied to glow-discharged QuantiFoil R1.2/1.3 Au 200 mesh grids, as previously reported[66]. The grids were blotted for 5 s and immersed in liquid ethane using Vitrobot Mark IV systems (FEI/Thermo Fisher) operated at 4 °C and 100% humidity. Grid observation and data collection were performed with a CRYO ARM™ 300II (JEOL) operated at 300 kV and equipped with a JEOL in-column Omega energy filter and a Gatan K3 BioQuantum detector. Movies were automatically collected using SerialEM software[65]. The data collection parameters are summarized in Supplementary Table 2.

## EM image processing

Movies were aligned using beam-induced motion correction implemented in RELION v3.1 and 4.0[67,68], and contrast transfer function parameters were estimated using Patch-based CTF estimation in CryoSPARC[69]. The following image processing was mainly performed by using CryoSPARC. Bayesian polishing was performed by RELION v3.1 and 4.0 with the help of UCSF pyem tool[70] for data conversion from CryoSPARC to Relion.

For the sample of ERGIC-53 in complex with MCFD2, 6345 movies were collected. For analysis of the head and TM regions, blob-based autopicking and subsequent 2D classification were performed by using 800 movies to create templates for template-based autopicking in CryoSPARC. A total of 2,132,031 particles were automatically picked by the template picker in CryoSPARC and extracted at a pixel size of 3.32 Å followed by 2D classification. Good classes with clear 2D average images of the head region were selected and subjected to ab initio modeling and subsequent heterogeneous refinements. Particles belonging to forms A and B were re-extracted at a pixel size of 0.996 Å and subjected to non-uniform (NU) refinement[71]. The refined particles were subjected to Bayesian polishing, and then NU refinement with global CTF and particle defocus refinement. Consequently, the final maps of forms A and B were refined to 2.53 and 2.59 Å resolution, respectively. The maps were sharpened and locally filtered based on the local resolution in CryoSPARC.

For its full-length particle analysis, full-length ERGIC-53 particles were manually picked from 50 micrographs. Particles showing full-length-like structures were selected from 2D classification and subjected to Topaz training[40]. A trained model of Topaz was further trained by repeating autopicking by Topaz and 2D classification using 50, 100, and 600 micrographs. Subsequently, full-length particles were picked by Topaz from all micrographs, and extracted at a pixel size of 2.49 Å. 2D classification generated clear 2D average images of full-length ERGIC-53 particles. Particles belonging to the straight conformation were selected and subjected to homogeneous refinement and local refinement. The final map was refined to 6.8 Å resolution.

For the ERGIC-53ΔH34-MCFD2 complex, 6175 movies were collected, and CTF parameters were estimated with CTFFIND4. Particles were initially picked using Laplacian-of-Gaussian-(LoG) based autopicking in RELION 4 to create 2D templates. A total of 127,9058 particles were picked and extracted at a pixel size of 3.142 Å and subjected to 2D classification. Good 2D classes were selected and subjected to ab initio modeling and subsequent heterogeneous refinements. The best particles were re-extracted at a pixel size of 1.3791 Å and subjected to homogenous refinement. The refined particles were subjected to Bayesian polishing, CTF refinement, defocus refinement, and homogenous refinement. The final map (consensus map) was reconstructed at 3.51 Å resolution.

To gain insight into the structural viability of all the obtained EM maps, the refined particles were subjected to 3D variability analysis in CryoSPARC[42]. In the head region analysis, particles were further classified by cluster analysis in 3DVA (12 classes), resulting in the determination of four substate structures at 3.3–3.4 Å resolution.

### Model building, refinement, and visualization

For the CRD region and MCFD2, the previously published crystal structure of the CRD and MCFD2 complex (PDB ID 3WNX) was docked in the EM map. The S-H1 and S-H2 regions were manually built by using Coot[72]. The models were refined with Servalcat[73] and phenix.real_space_refine[74] and further manually corrected with Coot. For the building of the full-length model, partial tetrameric models of S-H3-SH4, and S-H4-TM were predicted by using Colabfold with AF2 multimer[75]. Then partial models were assembled and manually docked in the EM map. The model was further refined with phenix_real_space refined as rigid bodies. Structural figures were prepared with PyMOL (http://www.pymol.org) and Chimera X[76]. Sequence alignment was performed and visualized with Jalview[77]. Prediction and analysis of coiled-coils was performed with Deepcoil2 and SamCC in MPI bioinformatics Toolkit[78].

### Secretion assay

HEK293T WT or ERGIC-53 KO cells ($6 \times 10^5$ cells) were plated on six-well plates and transfected with 100 ng of the plasmid expressing FV-HiBiT /AAT-HiBiT and 200 ng of the plasmid expressing ERGIC-53 WT or each of its mutant. After 24 h incubation, the medium was removed, and cells were washed with Expi293 medium and incubated in 1 ml of Expi293 medium for 4 h. In the $Zn^{2+}$ supplement conditions, 10 μM $ZnCl_2$ was added in the medium. After harvest of the conditioned mediums (CMs), the remained cells were washed with PBS and lysed in a lysis buffer (PBS, 1% Triton (Thermo Fisher), and protease inhibitor cocktail (Nakarai Tesque), and then briefly sonicated. The total volume of the cell lysates was adjusted to the same volume as the CMs. The amounts of secreted and intracellular HiBiT-tagged FV/AAT proteins were quantified by using the Nano-Glo HiBiT Extracellular Detection System or Nano-Glo HiBiT lytic assay system, respectively. Luminescence from HiBiT-tagged proteins was measured with GloMax Navigator (Promega). Luminescence from CMs and lysates of nontransfected cells were used for background correction.

Secreted and intracellular HiBiT-tagged FV or AAT proteins, as well as expressed ERGIC-53 were also detected by Western blotting with the Nano-Glo® HiBiT Blotting System (LgBiT protein) and anti-Factor V (Abcam, ab108614, 1:3300) or anti-LMAN1 (Abcam, ab125006, 1:5000) antibodies. Total proteins were detected by stain-free technology (Bio-Rad). The experiments were performed independently at least three times. Statistical analysis was performed with GraphPad prism 9.3, using a two-tailed unpaired $t$-test for the comparison of two groups or one-way ANOVA followed by Dunnett's test for the other comparisons.

### Reporting summary

Further information on research design is available in the Nature Portfolio Reporting Summary linked to this article.

## Data availability

The cryo-EM density maps have been deposited in the Electron Microscopy Data Bank (EMDB) under accession codes EMD-36467 (form A), EMD-36468 (form B), EMD-36469 (substate A), EMD-36470 (substate B), EMD-36471 (substate C), EMD-36472 (substate D), EMD-36479 (full length) and EMD-36482 (the ΔH34 mutant). The atomic coordinates have been deposited in the Protein Data Bank under accessions 8JP4 (form A), 8JP5 (form B), 8JP6 (substate A), 8JP7 (substate B), 8JP8 (substate C), 8JP9 (substate D), and 8JPG (full length). The following atomic coordinates were used in this study: 3WNX (the crystal structures of the CRD and MCFD2 complex), and 4YGE (Zn-free MCFD2 structure). Raw movies have been deposited in EMPIAR-11645 (the ΔH34 mutant) and EMPIAR-11646 (full-length ERGIC-53). Source data are provided with this paper.

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

## Acknowledgements

We would like to thank staff scientists at the University of Tokyo's cryo-EM facility, especially, A. Tsutsumi, K, Kobayashi, H. Yanagisawa, M. Kikkawa, and R. Danev; T. We also thank T. Yokoyama, K. Nanatani, J. Inoue, S. Koshiba, and M. Yamamoto for the management of the cryo-EM facility at Advanced Research Center for Innovations in Next-Generation Medicine in Tohoku University; and Y. Amagai and M. Inoue for help in culturing cells and cell biology experiments. This work was supported by Grants-in-Aid for Scientific Research (C) (JP18K06075 to S.W.), Grant-in-Aid for Transformative Research Areas (JP21H05253 to K.I. and S.W.), Grant-in-Aid for Challenging Research (Exploratory) (JP23K18193 to S.W.) from the MEXT of Japan, a research funding from AMED-CREST (21gm1410006h0001) to K.I., a research grant from the Naito foundation (to S.W.), a research grant from Mochida memorial foundation for medical and pharmaceutical research (to S.W.), a research grant from AMED (JP22gm6410026 to Y.K.), and Platform Project for Supporting Drug Discovery and Life Science Research (Basis for Supporting Innovative Drug Discovery and Life Science Research (BINDS)) from AMED under grant number JP21am0101115 (support number 1025) JP21am0101071 (support number 2078), JP23ama121012 (support number 5756), JP21am0101095 and JP23ama121038.

## Author contributions

S.W. designed the research and performed almost all experiments, cryo-EM data collection at Tohoku University, image processing, and model building and refinement. Y.K. performed grid preparation/optimization and cryo-EM data collection at the University of Tokyo under the supervision of O.N. K.Y. performed SEC-MALS/SAXS analysis at KEK under the supervision of N.S. M.I. performed plasmid construction and secretion assays. S.W. and K.I. wrote the manuscript with the help of all other authors. S.W. and K.I. supervised this work.

## Competing interests

O.N. is a co-founder, board member, and scientific advisor for Curreio. The remaining authors declare no competing interests.
