## [Peer Review File · Nature Communications]

Structure of full-length ERGIC-53 in complex with MCFD2 for cargo transportReviewer #1 (Remarks to the Author):

This manuscript describes a thorough structural analysis of the ER-to-Golgi cargo receptor ERGIC-53, a transmembrane protein that extends about 300 Å off the surface of the membrane. A cryo-EM structure of the full-length protein complex was used to definitively identify its oligomerization state and symmetry, correcting previous misconceptions from the literature. A zinc binding site was identified, inspiring a model for zinc-mediated regulation of cargo binding. The experiments are described clearly and convincingly, the integration of the structural information with possible functional implications is engaging, the text is for the most part polished and elegant, and the figures are clear and informative. In short, it was a pleasure to read and review this manuscript.

Addressing a few very minor textual issues could perhaps improve the manuscript even further:

Line 80- "demonstrated" means "show the existence or truth of," but since ERGIC-53 turned out not to be a hexamer, it could not have been demonstrated that it was a hexamer. "Proposed" might be a better word choice.

Line 110- It is not technically correct to say that the bands on the SDS-PAGE were presumed "to be" the hexamers, since the ~110 kDa band is a dissociation product. Better to say that the bands were presumed "to result from" the covalent and non-covalent hexamers.

Line 115- the term "heterotetramer" brings to mind a 4-subunit complex composed of different kinds of subunit (like hemoglobin is a heterotetramer of alpha and beta subunits). Perhaps just say, "...homotetramer of ERGIC-53 alone (221.4 kDa) and a complex of four ERGIC-53 molecules with four MCFD2 molecules (227 kDa)." There is no need for "respectively" in this sentence.

Line 117- it should be "and each protomer" rather than "and its each protomer"

Line 325- it is difficult to tell from the inset in Figure 3a that there is "extra" density. Perhaps change to, "The geometry of the histidine cluster and the presence of density at the center of the cluster are consistent with coordination of a metal, most likely a zinc ion."

Line 449- what is meant by "relevant cells"?

Line 623- should be "colors" instead of "color"

Line 660- the symbols in the plot are not just dots. There are triangles, etc.

Line 758- should it be "were loaded"?

Line 864- there is a typo in cryoSPARC

Line 899- "at biologically independently at"?

Reviewer #2 (Remarks to the Author):

In this manuscript, the authors report the three-dimensional structure of the cargo receptor complex formed between ERGIC-53 (also known as LMAN1) and MCFD2, as revealed by cryo-electron microscopy (cryo-EM). While previous structural studies focused on the carbohydrate recognition domain (CRD) of ERGIC-53, this research, for the first time, elucidates the structure of the full-length molecule—a significant achievement. The cryo-EM analysis was thorough, and the obtained three-dimensional structures are carefully described, including their dynamic aspects. However, more careful

consideration seems necessary for the discussions based on the resolved structures. The following points of concern are raised:

While ERGIC-53 was previously assumed to be a hexamer, the structure revealed in this study indicates a tetramer. Although this observation is intriguing, the oligomeric state of ERGIC-53 may vary with cell types or environmental conditions, as exemplified by yeast homologs of ERGIC-53, Emp46p and Emp47p, which have been reported to undergo a pH-dependent transition in their oligomeric state.

Since the CRD of ERGIC-53 is monomeric, it is likely that interactions within the stalk region govern its oligomeric state. In connection with the above point, a more detailed description of intermolecular interactions in the stalk region is necessary.

ERGIC-53 has a homologous protein, ERGL. Yet, this study did not discuss the possible oligomeric state of ERGL and the possibility of its heterooligomer formation with ERGIC-53 based on the obtained structure. Also, the presence of ERGL could be considered in the interpretation of functional analyses using ERGIC-53 KO cells.

The authors propose an intriguing hypothesis that Zn^{2+} binding to the N-terminal segment of MCFD2 can control cargo binding. It is not clearly described whether complex formation with ERGIC-53 is a prerequisite for MCFD2 to bind Zn^{2+} . The impact of Zn^{2+} binding on the conformational state of MCFD2, inhibiting ligand binding, could possibly be examined using NMR spectroscopy.

The authors also suggest the possibility that the assembled CRDs disassemble upon binding to glycan ligands. However, it is regrettable that this remains speculative. This hypothesis can be easily tested using SAXS or other techniques.

Reviewer #1 (Remarks to the Author):

This manuscript describes a thorough structural analysis of the ER-to-Golgi cargo receptor ERGIC-53, a transmembrane protein that extends about 300 Å off the surface of the membrane. A cryo-EM structure of the full-length protein complex was used to definitively identify its oligomerization state and symmetry, correcting previous misconceptions from the literature. A zinc binding site was identified, inspiring a model for zinc-mediated regulation of cargo binding. The experiments are described clearly and convincingly, the integration of the structural information with possible functional implications is engaging, the text is for the most part polished and elegant, and the figures are clear and informative. In short, it was a pleasure to read and review this manuscript. Addressing a few very minor textual issues could perhaps improve the manuscript even further:

Response: We thank the reviewer for his/her positive evaluation and valuable comments to our work. Our point-by-point responses are described below.

Comment #1: Line 80- “demonstrated” means “show the existence or truth of,” but since ERGIC-53 turned out not to be a hexamer, it could not have been demonstrated that it was a hexamer. “Proposed” might be a better word choice.

Response: According to this suggestion, we revised the sentence as follows.

“A subsequent study by Neve et al., however, proposed that ERGIC-53 existed in two forms, a covalent hexamer and a noncovalent hexamer” (line 77, in the revised manuscript).

Comment #2: Line 110- It is not technically correct to say that the bands on the SDS-PAGE were presumed “to be” the hexamers, since the ~110 kDa band is a dissociation product. Better to say that the bands were presumed “to result from” the covalent and non-covalent hexamers.

Response: According to this suggestion, we revised the sentence as follows.

“at ~300 kDa and ~110 kDa, which were presumed to result from the covalent and noncovalent hexamers of ERGIC-53, respectively.” (line 109, in the revised manuscript)

Comment #3: Line 115- the term “heterotetramer” brings to mind a 4-subunit complex composed of different kinds of subunit (like hemoglobin is a heterotetramer of alpha and beta subunits). Perhaps just say, “...homotetramer of ERGIC-53 alone (221.4 kDa) and a complex of four ERGIC-53 molecules with four MCFD2 molecules (227 kDa).” There is no need for “respectively” in this sentence.

Response: According to this suggestion, we revised the sentence as follows.

“The determined molecular masses correspond to a homotetramer of ERGIC-53 alone (221.4 kDa) and a complex of four ERGIC-53 protomers with four MCFD2 molecules (277 kDa)” (lines 112-114, in the revised manuscript)

Comment #4: Line 117- it should be “and each protomer” rather than “and its each protomer”

Response: According to this suggestion, we revised the sentence as follows.

“and each protomer forms a 1:1 complex with MCFD2.” (line 116, in the revised manuscript)

Comment #5: Line 325- it is difficult to tell from the inset in Figure 3a that there is “extra” density. Perhaps change to, “The geometry of the histidine cluster and the presence of density at the center of the cluster are consistent with coordination of a metal, most likely a zinc ion.”

Our response: According to this suggestion, we revised the sentence as follows.

“Notably, the geometry of the histidine cluster and the presence of density at the center of the cluster are consistent with coordination of a metal, most likely a zinc ion (Fig. 3a inset).” (lines 332-334, in the revised manuscript).

Comment #6: Line 449- what is meant by “relevant cells”?

Response: In response to this comment, we modified the indicated words as follows.

“a recent study reported that in dendritic cells and their related cells such as airway epithelial cells” (line 487-488, in the revised manuscript).

Comment #7: Line 623- should be “colors” instead of “color”

Response: We corrected the word (line 994, in the revised manuscript).

Comment #8: Line 660- the symbols in the plot are not just dots. There are triangles, etc.

Response: According to this suggestion, we revised the sentence as follows.

“Each symbol represents the individual data point.” (line 1035, in the revised manuscript)

Comment #9: Line 758- should it be “were loaded”?

Response: we corrected this typo (line 645 in the revised manuscript)

Comment #10: Line 864- there is a typo in cryoSPARC

Response: we corrected this typo. (Lines 249, 680,701, in the revised manuscript).

Comment #11: Line 899- “at biologically independently at”?

Response: we removed this phrase to get the following sentence.

“The experiments were performed independently at least three times” (line 737, in the revised manuscript)

Reviewer #2 (Remarks to the Author):

In this manuscript, the authors report the three-dimensional structure of the cargo receptor complex formed between ERGIC-53 (also known as LMAN1) and MCFD2, as revealed by cryo-electron microscopy (cryo-EM). While previous structural studies focused on the carbohydrate recognition domain (CRD) of ERGIC-53, this research, for the first time, elucidates the structure of the full-length molecule—a significant achievement. The cryo-EM analysis was thorough, and the obtained three-dimensional structures are carefully described, including their dynamic aspects. However, more careful consideration seems necessary for the discussions based on the resolved structures. The following points of concern are raised:

Response: We thank the reviewer for his/her positive evaluation and constructive comments to our work. Our point-by-point responses are described below.

Comment #1: While ERGIC-53 was previously assumed to be a hexamer, the structure revealed in this study indicates a tetramer. Although this observation is intriguing, the oligomeric state of ERGIC-53 may vary with cell types or environmental conditions, as exemplified by yeast homologs of ERGIC-53, Emp46p and Emp47p, which have been reported to undergo a pH-dependent transition in their oligomeric state.

Response: Thank you for raising the interesting possibility regarding the oligomeric state of ERGIC-53. Indeed, previous studies reported that the heterotetramer complex of Emp46p and Emp47p was stable at pH6~7, but could be dissociated at more acidic pH (at pH5 or lower), where a glutamate residue (Glu306) of Emp46p was suggested to function as a pH sensor to regulate its oligomeric state (Kato et al., JB 2018). However, we found that no charged residues are located in the hydrophobic core of the coiled-coil. The stalk helices of ERGIC-53 are much longer than those of Emp46p/Emp47p, suggesting tighter interactions between the coiled-coils in ERGIC-53. Furthermore, unlike Emp46p and Emp47p, the ERGIC-53 tetramer is stabilized by intermolecular disulfide bonds. Thus, the oligomeric state of ERGIC-53 is likely robust under wide environmental conditions.

To examine the pH dependency of the ERGIC-53 tetramer, we purified ERGIC-53 from cells throughout at pH7.2 (pH at ER) and pH 6.5 (pH at Golgi) and characterized its oligomeric state under both pH conditions by SEC (please see the result attached below, and a new supplementary Fig 2). SEC analysis showed that ERGIC-53 was eluted with almost the same elution volume at pH7.2 and pH6.5, suggesting that the tetrameric state of ERGIC-53 is stable in this range of pH. The broad peaks observed at the larger elution volume (16-18 ml) are likely ascribed to the nucleotide contamination, because no protein bands were observed for these elution fractions in the SDS-PAGE analysis (right panel).

Based on the above results, the oligomeric state of ERGIC-53 is likely retained during the cycling between the ER and ERGIC/Golgi. Although our results cannot exclude the possibility that different cell types may express different oligomeric states of ERGIC-53, this will be an issue to be explored in the future study.

In the revised manuscript, thus, we have added a new supplementary figure 2 and additionally mentioned about the different structural features of the stalk helix domain between ERGIC-53 and Emp46p/Emp47p and discussed the stability of the ERGIC-53 tetramer in the early secretory pathway (lines 114-115 and 405-419 in the revised manuscript).

Comment #2: Since the CRD of ERGIC-53 is monomeric, it is likely that interactions within the stalk region govern its oligomeric state. In connection with

the above point, a more detailed description of intermolecular interactions in the stalk region is necessary.

Response: According to this suggestion, we prepared a new figure showing the details of the intermolecular interactions in the stalk helix domain (Supplementary Fig. 5, in the revised manuscript). The S-H2 segment forms a typical parallel four-helix coiled-coil. Hydrophobic residues are located at the heptad positions **a** and **d** of the coiled-coil, forming the hydrophobic core within it (Supplementary Figs. 4b,c,d,e). A polar residue Gln338 is also located at the position “**a2**”, where a water molecule or some ion is bound to stabilize the polar interactions (Supplementary Fig. 4f).

We added the above structural information in the revised manuscript (lines 162-166, in the revised manuscript)

Comment #3: ERGIC-53 has a homologous protein, ERGL. Yet, this study did not discuss the possible oligomeric state of ERGL and the possibility of its heterooligomer formation with ERGIC-53 based on the obtained structure. Also, the presence of ERGL could be considered in the interpretation of functional analyses using ERGIC-53 KO cells.

Response: Thank you for providing your insights into ERGL. We used HEK293T-derived ERGIC-53 KO cells for the functional analysis (Fig. 4). Previous proteomic analysis of eleven common cell lines including the HEK293 cell line reported that ERGL was not expressed at a detectable level (Geiger T et al., *Mol. & Cell. Proteo.* 2012), suggesting that the contribution of endogenous ERGL was negligible in our rescue experiment. The putative stalk helix domain of an ERGL protomer is also predicted to contain three coiled-coil helices, although the domain has a low sequence similarity to that of ERGIC-53 (38% similarity). It is thus possible that ERGL exists as a tetramer (New Supplementary figure 14a). Interestingly, the program “AlphaFold2 multimer” predicted possible formation of a heterotetramer structure of ERGIC-53 and ERGL via three sets of four-helix coiled coils (S-H2 to S-H4) (new supple fig 14b). Thus, ERGL may form a heterotetramer with ERGIC-53 to function as an additional or auxiliary cargo receptor in some tissues and cells.

We added the above discussion and a new supplementary figure (new supplementary Fig. 14) showing the predicted tetrameric structure of ERGL and ERGIC-53 in the revised manuscript (lines 420-431, in the revised manuscript).

Comment #4: The authors propose an intriguing hypothesis that Zn²⁺ binding to the N-terminal segment of MCFD2 can control cargo binding. It is not clearly described whether complex formation with ERGIC-53 is a prerequisite for MCFD2 to bind Zn²⁺. The impact of Zn²⁺ binding on the conformational state of MCFD2, inhibiting ligand binding, could possibly be examined using NMR spectroscopy.

Response: Our ITC analysis showed that MCFD2 alone has a zinc binding ability (Fig. 3c). Consistently, the present cryo-EM analysis of the ERGIC-53-MCFD2 complex revealed that MCFD2 purified from *E. coli* cells bound Zn²⁺ even without external addition of Zn²⁺. Based on these observations, it is unlikely that complex formation with ERGIC-53 is prerequisite for MCFD2 to bind zinc. To highlight Zn²⁺-induced conformational changes in MCFD2, we prepared a new figure showing the superposition of the Zn²⁺-bound and –unbound forms of MCFD2 (new Fig. 3b). The new figure shows that no large conformational changes are induced upon zinc binding, except for its N-terminal segment. (line 324-325 in the revised manuscript).

We added the above discussion with the insertion of the following sentences to the revised manuscript (lines 460-466, in the revised manuscript).

“The ITC analysis for Zn²⁺ binding to MCFD2 and purification of MCFD2 in metal ion (Zn²⁺)-bound form from E. coli cells suggest that Zn²⁺ binding ability of MCFD2 does not rely on the complex formation with ERGIC-53. While overall structure of MCFD2 is not largely altered upon Zn²⁺ binding, its cargo-binding site is masked by its N-terminal helix in the present cryo-EM structure, and this local structure is probably stabilized by the Zn²⁺-bound His cluster (Fig. 3a).”

Comment #5: The authors also suggest the possibility that the assembled CRDs disassemble upon binding to glycan ligands. However, it is regrettable that this

remains speculative. This hypothesis can be easily tested using SAXS or other techniques.

Response:

We agree that our proposed model of the CRD assembly/disassembly regulated by glycan ligands is still speculative. To prove the model solidly, detailed structural information on the ERGIC-53-MCFD2-cargo (with glycan ligand) complex would be necessary. This is absolutely the next important project to be challenged. Given that, we toned down our statement regarding the model by revising the related sentences as follows (Lines 445-447 in the revised manuscript).

“Thus, the present CRD assembled structure possibly represents a post-cargo release state of the ERGIC-53-MCFD2 complex. Cargo binding may alter relative positions of the CRDs to capture a target cargo protein in concert with MCFD2.”

Reviewer #1 (Remarks to the Author):

The manuscript was excellent to begin with, and the authors have further improved it in response to the reviewers' comments.

Reviewer #2 (Remarks to the Author):

The authors have effectively addressed the concerns I raised, leading to a noticeable enhancement in the overall quality of the paper. Regarding my comment on Zn²⁺ binding, I suggested that utilizing NMR could offer valuable insights to validate whether conformational rearrangement within the N-terminal disordered region indeed takes place during metal coordination. However, it should be noted that this aspect remains a topic for future consideration. Nevertheless, I am positive about the publication of this revised manuscript regardless.